

# An investigation of the appearance of a long range nuclear potential in ultra low energy nuclear synthesis

**Shinsho Oryu** [1*]**, Takashi Watanabe** [1]**, Yasuhisa Hiratsuka** [2] **and Masayuki Takeda** [3]

**1** Department of Physics, Faculty of Science and Technology,
Tokyo University of Science, Noda, Chiba 278-8510, Japan
**2** Preparation School of HLF Ltd, Kiryu, Gunma 376-0021, Japan
**3** Department of Information Science, Faculty of Science and Technology,
Tokyo University of Science, Noda, Chiba 278-8510, Japan

⋆ oryu@rs.noda.tus.ac.jp

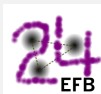

## Abstract

A new potential, the *general particle transfer potential*: so called "GPT" potential was represented in the previous paper which indicates a Yukawa-type potential for shorter range, but a $1/r^n$-type potential for longer range where $n = 2$ includes the Efimov-like potential in the hadron system. In order to confirm the existence of a GPT potential, we investigate the possibility of Cs+2d→La reaction on the three-ion quasi-molecule $CsD_2$ which is covered by twelve Pd or a $CsD_2Pd_{12}$-cluster, where the three-body bound states and wave functions for D-Cs-D molecular and d-Cs-d nuclear systems are calculated. We obtain an approximate E2-transition from the molecular states to the nuclear states. The transition ratio between the short range nuclear potential with the $1/r^2$-type long range potential and without long range potential is $W_{i \to f}^{E2';L}/W_{i \to f}^{E2';S} \approx 10^8$. If the reaction Cs+2D→La is experimentally observed, then the existence of the GPT potential could be confirmed.


## 1 Introduction

Based on three-body kinematics, we proposed a *general particle transfer* (GPT) potential which appears, not only at the three-body break-up threshold (3BT) but also at the quasi two-body threshold (Q2T) [1] [2]. The "GPT-potential" is a Yukawa-type potential for shorter range, but a $1/r^n$-type potential for longer range, where $n = 2$ includes the Efimov-like potential [3] - [5]. We briefly review the GPT potential in section 2. The GPT potential could generally exist in many systems, from atomic-molecular physics to hadron physics, especially in the threshold behaviors where the two-body interaction becomes very small.

In this paper, we explore a subject named the "ultra low energy nuclear synthesis" in section 3. In the traditional nuclear fusion approach, the most serious problem is whether the incident energy is sufficient to penetrate the Coulomb barrier, because the reaction is usually started in the free field. We point out that the Cs-d penetration in the quasi-molecule, which is developed in the Pd-crystal, could easily occur [6], since the energy levels of such a quasi-molecule could exist close to the top of the Cs-d Coulomb barrier or over the barrier in the "Pd-cage" which is a wall by $Pd_{12}$. Experimental results were published in 2002 by Iwamura et al. [7], and Hioki et al. in 2013 [8]. The most critical point, whether the nuclear synthesis in their experiments did occur, will be discussed in our theoretical view in section 4.

## 2 The GPT potential

In the three-body AGS equations, the coincidence of singularities in the kernel arises between the particle transfer term and the propagator of the two-body sub-system [9] which brings about a serious problem at the 3BT. On the other hand, in the quasi two-body scattering problem, where the two-body subsystem has a bound state in the three-body problem, the kernel also diverges in the particle transfer diagram and the two-body Green's function at the Q2T. It should be noted that the 3BT corresponds to the case of two-body zero binding energy in the Q2T. Therefore, the Q2T is more general than the 3BT. Such a coincidence singularity is also found in the two-body Lippmann-Schwinger equation with the Coulomb potential which gives rise to a serious problem with the Green's function at the on-shell threshold [10]- [12]. In the Coulomb problem, such as the electron-proton system, the potential is given by

$$V(r) = -e^2/r. \tag{1}$$

Efimov pointed out that the three-body bound states with the two-body scattering length of $a_0 \to \infty$ accumulate on the 3BT [4] [5]. Historically, Nicholson pointed out in 1962 that such a scattering length is exactly given by another type of attractive long range potential in the two-body Lippmann-Schwinger equation [3],

$$V(r) = -\alpha/r^2, \tag{2}$$

with a proper constant $\alpha > 0$. Therefore, one should keep in mind that such long range potentials are closely related with the coincidence of singularities in the kernel of the two- and three-body scattering equations.

In 2012, the GPT potential was proposed not only at the 3BT but also at the Q2T in the three-body problem [1] [2]. As mentioned above, since the Q2T singularity is more practical in the hadron system than the 3BT case where the "finite value of the scattering length" for the two-body potential is possible with the binding energy $\epsilon_B \neq 0$, although at the 3BT, Efimov persists in the two-body "infinite value of the scattering length" [4].

Let us recall our theory briefly at the Q2T. The Born term of the AGS equation is given in terms of two-body form factors with different channels $g_\alpha(\vec{p})$, $g_\beta(\vec{p}')$,

$$Z_{\alpha\beta}(\vec{q}, \vec{q}'; E) = \frac{g_\alpha(\vec{p}) g_\beta(\vec{p}')(1 - \delta_{\alpha\beta})}{E - q^2/2\mu - p^2/2\nu} \to \frac{g_\alpha(\vec{p}) g_\beta(\vec{p}')(1 - \delta_{\alpha\beta})}{E - q^2/2\mu + \epsilon_B}, \tag{3}$$

where two-body momenta $\vec{p}$, $\vec{p}'$ are represented by the linear combination of $\vec{q}$ and $\vec{q}'$ with masses. $\nu$ is the two-body reduced mass between two particles, and $\mu$ represents the reduced mass between a spectator and the two-body masses, respectively. The three-body free energy is given by $E$.

For the Q2T: $E_{cm} \equiv E + \epsilon_B = 0$, $\vec{q} = 0$, and Eq.(3) becomes

$$Z_{\alpha\beta}(\vec{q}, \vec{q}'; E) \to \infty. \tag{4}$$

On the other hand, at the Q2T, the two-body bound state becomes $\epsilon_B \neq 0$, and therefore, the Efimov criterion : the scattering length $a_0 \to \infty$ is not satisfied anymore. The propagator at the Q2T becomes with the on-shell relation $z = E - q^2/2\mu \ (\leq 0)$,

$$\tau_B(z) \ = \ \frac{f(z)}{\epsilon_B + z} = \frac{f(z)}{(\epsilon_B + E) - q^2/2\mu} \to \infty. \tag{5}$$

Therefore, Eq.(4) and Eq.(5) indicate the "singularity coincidence" at the Q2T, where one could imagine "an existence of long range potential".

Just below the Q2T ($E_{cm} \leq 0$), the AGS-Born term [9] becomes with $\vec{p} = \vec{p}' = 0$,

$$Z_{\alpha\beta}(\vec{q}, \vec{q}'; E) = \frac{g_\alpha(0)g_\beta(0)(1 - \delta_{\alpha\beta})}{-|E_{cm}| - q^2/2\mu} = -\frac{C_{\alpha\beta}}{q^2 + \sigma^2}, \tag{6}$$

where $C_{\alpha\beta} = 2\mu g_\alpha(0)g_\beta(0)(1 - \delta_{\alpha\beta})$, and $\sigma = \sqrt{2\mu|E_{cm}|}$. Therefore, the Fourier transformation of this energy dependent potential becomes,

$$\mathcal{F}[Z_{\alpha\beta}(\vec{q}, \vec{q}'; E)] = -\mathcal{F}\Big[\frac{C_{\alpha\beta}}{q^2 + \sigma^2}\Big] = V_0 \frac{e^{-\sigma r}}{r}, \tag{7}$$

with $V_0 < 0$. The $r$-space potential is a kind of Yukawa potential, but energy dependent. For $\sigma = 0$ or $E_{cm} = 0$, it becomes the Coulomb potential (or the gravitational potential); therefore, our AGS equation is essentially the same equation as the Coulomb Lippmann-Schwinger equation such as in electron-proton scattering except for the coupling constant [9]. In order to solve the eigenvalue equation with the energy dependent potential of Eq.(7), we have to solve it consistently with the two energies which are seen in the potential and in the eigenvalue. However, the method is very complicated and hard to obtain with good accuracy. Therefore, we introduced in ref. [1], an energy average by using a probability density function with respect to the possible energy range, which also represents effects of the structure or the form factors of the composite particles,

$$P_\sigma = \frac{\sigma^{2\gamma+1}e^{-a\sigma}}{\rho}, \tag{8}$$

with

$$\rho = \int_0^\infty \sigma^{2\gamma+1}e^{-a\sigma}d\sigma = \frac{\Gamma(2\gamma+2)}{a^{2\gamma+2}}, \tag{9}$$

where $e^{-a\sigma}$ is a damping factor with a "range parameter $a$" which should not be confused by the scattering length. By using the probability density function, the expectation value of the energy-dependent potential becomes energy independent. This transformation is called the "Euler integral of the second kind" with respect to Eq.(7), or a Laplace transformation with a weight function. Therefore, by using Eq.(8) and Eq.(9), Eq.(7) becomes,

$$\begin{aligned} \mathcal{L}\Big\{\mathcal{F}[Z_{\alpha,\beta}(\vec{q}, \vec{q}'; E)]\Big\} \ &= \ \mathcal{L}\Big\{\frac{V_0 e^{-\sigma r}}{r}\Big\} = \frac{V_0}{\rho}\int_0^\infty \sigma^{2\gamma+1}e^{-a\sigma}\frac{e^{-\sigma r}}{r}d\sigma \\ &= \ V_0 \frac{a^{2\gamma+2}}{r(r+a)^{2\gamma+2}}. \end{aligned} \tag{10}$$

Thus, the predicted GPT potential is obtained.

Table 1: The GPT potential $V_0 a^{2\gamma+2}/[r(r+a)^{2\gamma+2}]$ is illustrated, which is given by an energy average below the 3BT ($E = 0$, $\epsilon_B = 0$) and below the Q2T ($E = -\epsilon_B$, $\epsilon_B \neq 0$) with two-parameters $a$ and $\gamma$. The potential properties for the longer and shorter ranges are shown with respect to the parameter $\gamma$. $V_0(< 0)$, a potential depth which is analytically given by Eq.(7).

| $\gamma$ | short range potential $r \ll a$ | potential | long range potential $a \ll r$ |
|---|---|---|---|
| $-1$ | $V_0/r$ | $V_0/r$ | $V_0/r$ |
| $-1/2$ | $V_0 e^{-r/a}/r$ | $V_0 a/[r(r+a)]$ | $V_0 a/r^2$ |
| $0$ | $V_0 e^{-2r/a}/r$ | $V_0 a^2/[r(r+a)^2]$ | $V_0 a^2/r^3$ |
| $1/2$ | $V_0 e^{-3r/a}/r$ | $V_0 a^3/[r(r+a)^3]$ | $V_0 a^3/r^4$ |
| $1$ | $V_0 e^{-4r/a}/r$ | $V_0 a^4/[r(r+a)^4]$ | $V_0 a^4/r^5$ |
| $3/2$ | $V_0 e^{-5r/a}/r$ | $V_0 a^5/[r(r+a)^5]$ | $V_0 a^5/r^6$ |
| $2$ | $V_0 e^{-6r/a}/r$ | $V_0 a^6/[r(r+a)^6]$ | $V_0 a^6/r^7$ |
| $\cdot$ | $\cdots$ | $\cdots$ | $\cdots$ |
| $\cdot$ | $\cdots$ | $\cdots$ | $\cdots$ |

Therefore, it is seen that the GPT potential is a combination of a Yukawa-type potential at shorter range but $1/r^n$-type potential for longer range. $\gamma = -1$ gives an attractive Coulomb or the gravitational-type potential for any region which is given by zero mass particle transfer. On the other hand, $\gamma = -1/2$ means the Efimov-type potential for the longer range. One could imagine that the mass of the transfered particle could be very small compared to the total mass of the parent particles. Therefore, an interaction between Cs-atom and D-atom by electron transfer could be given by the $1/r^2$-type potential. While a one pion transfer potential between the Cs- and d-nuclei could be the $1/r^2$-type or $1/r^3$-type which should be added to the nuclear Cs-d Wood-Saxon potential.

# 3 An Investigation of the Long Range Force in the Cs(2d,$\gamma$)La Reaction

It is expected that many three-body systems such as unstable nuclei, nuclear halo systems and hypernuclei as well as nuclear forces could be affected by the GPT potential. Especially, the long range effect due to the GPT potential could appear in a kinematic region where the two-body potential becomes weak.

Recently, it was claimed that several incomprehensible phenomena were found in the condensed matter nuclear science field. In 2002, Iwamura et al. found that, when Cs is added on the surface of a Pd complex, $D_2$ (hydrogen molecule) gas permeation at 343K changes Cs to Pr which means that a reaction: Cs+2$D_2 \to$ Pr occurs [7]. They also found that a reaction: Sr+2$D_2 \to$ Mo also occurs. Furthermore, Hioki et al. confirmed the previous reactions in 2013 [8].

Usually, in the free field, it is known that the molecular state is stable, and never changes to the nuclear state, because the molecular state is energetically too far from the nuclear state and the wave functions could not overlap one another, and also the Coulomb barrier can not be penetrated at a ultra low energy compare to the usual nuclear fusion reactor.

However, in condensed matter, if and only if, some supplemental states generated by the long range nuclear potential could mediate a transition from the molecular state to nuclear states, the nuclear synthesis could occur as an electro-magnetic (EM) transition between both states. Finally, the molecular state: CsD$_4$ could change to a Pr-nucleus which could be accomplished

by the direct five-body transition $CsD_4 \to Pr$, or by the sequential reaction $CsD_2 \to La$ and $LaD_2 \to Pr$.

In this paper, we first investigate the three-body reaction problem: $CsD_2 \to La$ in the Pd complex instead of the five-body problem: $CsD_4 \to Pr$, and compare two cases with our long range potential plus nuclear potential and without the long range potential. Therefore, this reaction could be represented by $Cs(d_2,\gamma)La$ in nuclear reaction terminology in the $CsD_2Pd_{12}$ system with the shape of cub-octahedron [14], where $\gamma$ stands for an energetic photon. Although, the Coulomb barrier still exists in this problem, we should say again that the barrier is not in the free field but in the Pd cluster (or Pd-cage). For this reason, the energy levels can be calculated from the bottom of the ion-ion potential to the top of the Coulomb barrier. Therefore, the penetration problem is resolved in the quasi-molecule: $CsD_2$ in the $CsD_2Pd_{12}$ system, and $CsD_4$ in $CsD_4Pd_{12}$, and also $CsD_6$ in the system $CsD_6Pd_{12}$ [13], [14].

In order to investigate a very shallow nuclear state, we would like to study the D-Cs-D three-ion problem by a high precision three-body variational (HPV) method with 80 to 100 digits of accuracy based on a usual variational approach. For the nuclear potentials Cs-d, d-d, the WS potential is adopted. The repulsive Coulomb potentials for Cs-D, D-D, and (Cs/D)-Pd cage are taken into account, where the repulsive Coulomb (Cs/D)-Pd potentials are given by a one-body $V_c^{Pd}$ potential. We also adopted a three-cluster force to fit the ground state of La: $V_t$. The two- or three-body long range hadron interactions are introduced by the GPT potential which are given by a kind of "three-body Efimov potential" [4]. Some parameters of the potential are chosen to cancel the three-body force effects at the tail of the WS potential [14], where the so-called "three-body Efimov potential" should not be confused with the two-body potential which produces the infinite value of the two-body scattering length.

The WS potential is given for the $i^{th}$-N and $j^{th}$-N nuclear potential: $V_W^{N_i N_j}(r_{ij})$,

$$V_W^{N_i N_j}(r_{ij}) = V_{W0}^{N_i N_j}\Big[1 + \exp\Big(\frac{r_{ij} - R_W^{N_i N_j}}{a_W^{N_i N_j}}\Big)\Big]^{-1}, \tag{11}$$

with parameters, $V_{W0}^{Csd} = -79.30$MeV, $V_{W0}^{dd} = -27.57$MeV, $R_W^{Csd} = 10.21$fm, $R_W^{dd} = 1.49$fm, $a_W^{Csd} = 0.4$fm, $a_W^{dd} = 0.3$fm, respectively. Hereafter, notations $r_{ij} = |\vec{r}_i - \vec{r}_j| = |r_{ij}|$ and $r_i = |\vec{r}_i|$ are used without the vector sign. We adopted only the central part of the potential, and the non-central part of the WS potential has been omitted for the first time, because the highest nuclear energy level is not very sensitive to the non-central part. Furthermore, the EM transition from the molecular state to the highest nuclear energy level is very important for the nuclear synthesis. Therefore, we calculate only the S-state wave function in the beginning. The ion-ion Coulomb potential is

$$V_c^{N_i N_j}(r_{ij}) = \begin{cases} \dfrac{Z_i Z_j e^2}{8\pi R_c^{N_i N_j}}\Big[3 - \Big(\dfrac{r_{ij}}{R_c^{N_i N_j}}\Big)^2\Big] & \text{for} \quad r_{ij} \le R_c^{N_i N_j}, \\[3mm] \dfrac{Z_i Z_j e^2}{4\pi r_{ij}} & \text{for} \quad R_c^{N_i N_j} < r_{ij}, \end{cases} \tag{12}$$

with $R_c^{Csd} = 10.21$fm, and $R_c^{dd} = 1.49$fm.

The Pd-$N_i$ Coulomb barrier is represented by a one-body potential,

$$V_c^{PdN_i}(r_i) = V_{c0}^{Pd}\Big(\frac{r_i}{a_c^{Pd}}\Big)^{10}\exp\Big\{-\Big(\frac{r_i - a_c^{Pd}}{b_c^{Pd}}\Big)^2\Big\}, \tag{13}$$

with $V_{c0}^{Pd} = 1.0 \times 10^{-4}$MeV, $a_c^{Pd} = 5.0 \times 10^5$fm, and $b_c^{Pd} = 3.1623 \times 10^5$fm, respectively. These parameters for the Pd-cage indicate that the location of Pd is $1.57 \times 10^6$fm, and 2.73MeV

height, however, these values may vary.

In this paper, we adopt a three-cluster (or body) potential as,

$$V_t(\vec{r}_1, \vec{r}_2, \vec{r}_3) = V_{t0} \exp\left[-\left(\frac{\vec{r}_1 - \vec{r}_2}{a_t}\right)^2 - \left(\frac{\vec{r}_2 - \vec{r}_3}{a_t}\right)^2 - \left(\frac{\vec{r}_3 - \vec{r}_1}{a_t}\right)^2\right], \tag{14}$$

where $V_{t0} = 1800$MeV and $a_t = 3.0$fm are used to fit the ground state of La by adding the Coulomb force.

We obtain a very good fit to the experimental ground state energy $\mathcal{E}_0 = -32.3$MeV, the root mean square (rms) radius $R_0 = 6.25$fm, the highest excited energy $\mathcal{E}_{max} = -3.5134 \times 10^2$keV, for the usual nuclear potential by the HPV calculation. Therefore, the highest energy level is far from that of the usual molecular state.

On the other hand, in order to obtain shallow energy levels that could be close to the molecular states we adopt a kind of three-body long range potential with a $1/r_{ij}^n$ tail which was proposed as the GPT potential [1] [2] with a Lorentz form or given by the modified three-body Efimov potential [4]:

$$V_e(\vec{r}_1, \vec{r}_2, \vec{r}_3) = V_{e0} a_e^n \left[(\vec{r}_1 - \vec{r}_2)^n + (\vec{r}_2 - \vec{r}_3)^n + (\vec{r}_3 - \vec{r}_1)^n + a_e^n\right]^{-1}, \tag{15}$$

where we choose $n = 2$ in this paper. Parameters $V_{e0} = -80000$MeV, and $a_e = 5000$fm are taken, and also a supplemental $a_e^n$ is used to avoid divergence. It could be noticed that these parameters seem to be too large to compare to the parameters in the WS potential, however Eq.(15) does not change the usual nuclear potential in the shorter range together with the three-body potential.

The three-body Efimov potential with $n = 2$ in Eq.(15) is analogous with the AGS Born diagram for the $k$-particle transfer which could be classified into two types: A and B,

A) The three-body long range potentials between parent particles are represented by

$$V_e(\vec{r}_1, \vec{r}_2, \vec{r}_3) = V_{e0} a_e^2 / [r_{ij}^2 + a_e^2] \qquad \text{(for } \vec{r}_{ki} = 0 \text{ and } \vec{r}_{jk} = 0\text{)}, \tag{16}$$

$$= V_{e0} a_e^2 / [2r_{ij}^2 + a_e^2] \qquad \text{(for } \vec{r}_{ki} = 0 \text{ or } \vec{r}_{jk} = 0\text{)}. \tag{17}$$

B) The two-body long range potentials in the AGS Born term appears in the form factor which is represented by the long range potential and the wave function:
$|g^L(\vec{r}_{kj})\rangle = V^L(\vec{r}_{kj})|\Psi^L(\vec{r}_{kj})\rangle$. Therefore, Eq.(15) for $n = 2$ is,

$$V_e(\vec{r}_1, \vec{r}_2, \vec{r}_3) = V_{e0} a_e^2 / [2r_{kj}^2 + a_e^2] \propto V^L(\vec{r}_{kj}) \tag{18}$$

$$= V_{e0} a_e^2 / [2r_{ki}^2 + a_e^2] \propto V^L(\vec{r}_{ki}) \qquad \text{(for } \vec{r}_{ij} = 0 \text{ )}. \tag{19}$$

C) Putting $\vec{r}_k = (\vec{r}_i + \vec{r}_j)/2$ for the fixed $k$-particle, Eq.(15) for $n = 2$ becomes

$$V_e(\vec{r}_1, \vec{r}_2, \vec{r}_3) = V_{e0} a_e^2 / [3r_{ij}^2/2 + a_e^2]. \tag{20}$$

Eq.(20) represents another type of three-body long range potential of (A). The three-body force long range potential of Eq.(15) indicates the non-linearity of the GPT potential which gives not only two-body long range but also three-body long range.

The highest three-cluster (or -ion) nuclear level can be well represented by the S-wave or even central potential. While, the La ground state was adjusted by using the central two-cluster Cs-d and d-d potentials to obtain $-32.3$MeV with the three-cluster force in Eq.(14). We confirmed that the phenomenological method for the La ground state is not very sensitive to the shallow bound states.

The nuclear interaction usually appears in the region of $0 \le r < 10^3$fm, however, the molecular levels have been calculated in the region of $10^4$fm$\le r < 10^6$fm. Therefore, it was historically

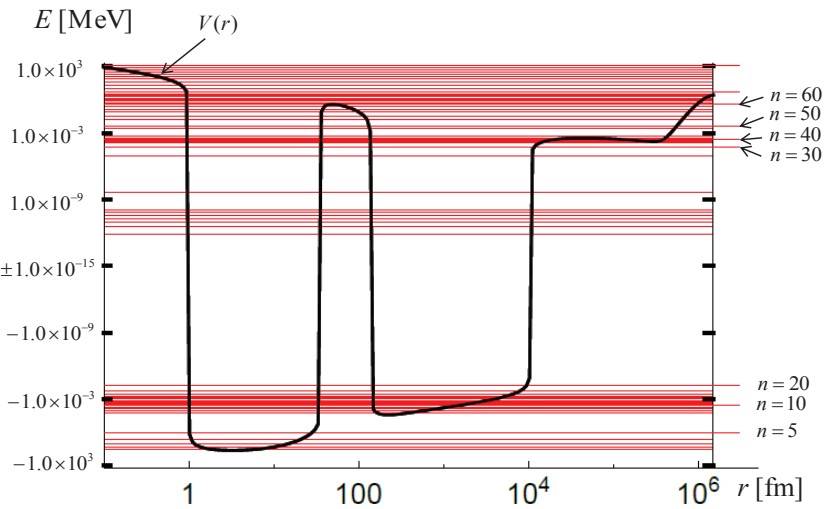

Figure 1: Three-body energy levels and potential "with the long range potential". The attractive potential part of the shorter range (fm region) indicates mainly the nuclear WS potential, and the next attractive part of $10^3$ fm region comes from mainly the GPT potential, and the final hollow in the $10^5$ fm region is the Coulomb attractive potential by the Cs ion and the Pd cage. $n$ is the principal quantum number. $1 \leq n \leq 5$ shows the La nuclear levels, $6 \leq n \leq 20$ are special levels by the GPT potential which belong to the nuclear excited states, and $21 \leq n \leq 60$ states represent the molecular levels for $CsD_2$, respectively.

thought that the nuclear state wave function and the molecular state wave function do not overlap to give rise to the EM transition. This means that the molecular state cannot transfer to the nuclear state. However, the binding energy and wave function of $CsD_2$ in the $Pd_{12}$-cage are completely different from the free system. We solved the three-ion system in the $Pd_{12}$-cage for the full region: $0 \leq r < \infty$ by the HPV method. In other words, for the nuclear and the molecular systems we can solve the eigen-equation on the basis of a common field in the full region with 80-100 figure accuracy.

A part of the E2 transition operator and the charge $Q_k$ of the $k$-th cluster is given by

$$o_k = Q_k(3z_k^2 - r^2) = Q_k r_k^2(3\cos^2 -1). \tag{21}$$

However, in our simple S-wave calculation, the value of $\sum_{k=1}^{3} < \Psi_f|o_k|\Psi_i >$ becomes zero by the initial and the final wave functions $\Psi_i$ and $\Psi_f$. Therefore, we replace $o_k$ by $Q_k r_k^2 = Z_k e r_k^2$.

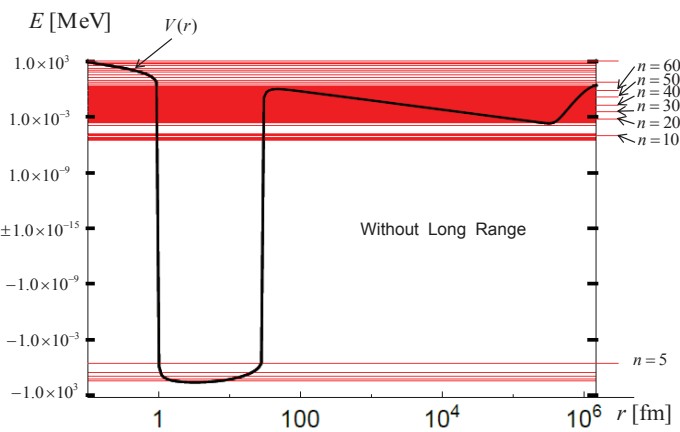

Figure 2: Energy levels and potential "without long range potential". The attractive potential part of the shorter range (fm region) indicates mainly the nuclear WS potential, and the next attractive part in the $10^5$fm region is the Coulomb attractive potential between the Cs ion and the Pd cage. $n$ is the principal quantum number. $1 \leq n \leq 5$ shows the La nuclear levels, and $6 \leq n \leq 60$ states represent the molecular levels for $CsD_2$, respectively.

The calculated results for the E2 transition is represented by

$$W_{i \to f}^{E2}(E_i, E_f) = \frac{1}{20} \frac{4(E_i - E_f)^5}{3\pi\varepsilon_0(h/2\pi)^6 c^5} \sum_{k=1}^{3} \Big| < \Psi_f | \frac{1}{2} Z_k e(3z_k^2 - r^2) | \Psi_i > \Big|^2 \quad (22)$$

$$\to W_{i \to f}^{E2'}(E_i, E_f) = \frac{1}{20} \frac{4(E_i - E_f)^5}{3\pi\varepsilon_0(h/2\pi)^6 c^5} \sum_{k=1}^{3} \Big| < \Psi_f | \frac{1}{2} Z_k e r_k^2 | \Psi_i > \Big|^2. \quad (23)$$

Therefore, the transition time is given by the inverse of $W_{i \to f}^{E2'}(E_i, E_f)$.

Finally, we obtained the ratio between $W_{i \to f}^{E2';L}(E_i, E_f)$ of the transition probability by the long range plus short range nuclear potential and $W_{i \to f}^{E2';S}(E_i, E_f)$ of the short range nuclear potential only,

$$\frac{W_{i \to f}^{E2';L}(E_i, E_f)}{W_{i \to f}^{E2';S}(E_i, E_f)} = \frac{1.5 \times 10^{16}/S}{1.1 \times 10^8/S} \approx 10^8, \quad (24)$$

with $S = cm^2$. The value of $W_{i \to f}^{E2';L}(E_i, E_f)$ is one order larger than the experimental value ($\sim 10^{15}/cm^2$) by Iwamura et al. for the $^{133}Cs + 4d \to ^{141}Pr$, although our calculated result indicates the reaction: $^{135}Cs + 2d \to ^{139}La$. Therefore, we conclude that the long range potential is essential to represent the experimental value.

## 4 Conclusion and Discussion

This work is based on our hypothesis of the long range GPT potential which is introduced by the three-body Faddeev method [15]. The GPT potential could emerge for the two-body form factor in the AGS Born term, because the two-body problem could be treated again by a certain three-body problem. Therefore, the long range effects in the three-body equation can not be written in terms of the sum of two-body long range potentials, but must be given by a three-body force-type or a nonlinear type potential. This is essentially predicted by Efimov in his first paper which we call the "long range three-body Efimov potential" based on the GPT potential, although several parameters of the potential are unfortunately not well defined.

In order to explore a possibility of the long range force, we discussed the nuclear synthesis at the ultra low energy. Evidence of the low energy nuclear synthesis was reported by Iwamura et al. in 2002 [7] and also in 2015 [16]. They measured the quantity of Cs and Pr by the XPS (X-ray photoelectron spectrometer) without taking out of the vacuum chamber, or in the Pd-cage where they measured unusual thermal release, but not observed the $\gamma$-ray emission. After the deuteron permeation, they etched the surface of Pd complex by using the $HNO_3$. The etched solutions or the ions without the Pd-cage were measured by ICP-MS (inductively coupled plasma mass spectrometry) etc., and finally the Pr element was confirmed by the mass. Finally, it should be mentioned that the most important improvement in our calculation method is that the whole region from 0.01fm to several nano meters in the three-body equation is solved in a straight forward way with a very accurate method providing 100 digits. Furthermore, we found that the wave functions oscillate very quickly inside the potential barrier instead of being damped exponentially. Therefore, our calculated wave functions give non-zero overlapping values between the molecular states and the proper nuclear states even for the case without the long range potential, not to mention for the case with the GPT potential which is accompanied by some additional states. In any case, our calculated result indicates that a different reaction rate should occur "without" the long range force and "with" the long range force, although we adopted only the S-wave calculation in this paper for simplicity. We believe that our calculation is the first serious work for the ultra low energy nuclear synthesis based upon the Faddeev approach using a long range potential [14].

## Acknowledgements

The authors would like to acknowledge valuable discussions with Drs. N. Watari, H. Kakigami, N. Hamada and Y. Fukumoto regarding the theoretical aspects of the molecular system, as well as I. Toyoda, and S. Tsuruga for sharing with us their experimental insight. We are indebted to MHI Innovation Accelerator LLC Co. Ltd. for significant financial support.

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
