# Peer review of "An investigation of the appearance of a long range nuclear potential in ultra low energy nuclear synthesis"

_SciPost Physics Proceedings, doi:SciPost Phys. Proc. 3, 050 (2020)_

## Round 1 · Referee Report · LAURO TOMIO · 2019-12-4

Strengths

The submitted work is enough interesting and appropriate to a few-body contribution in the SciPost Physics Proceedings. The authors, by following previous work are giving some more details on a proposal of a "general particle transfer" potential (named as GPT potential), applied to atomic-molecular systems.

Weaknesses

1-) A text revision is necessary, considering my comments and recommendations in the full report.
2-) Formalism should be corrected
and re-checked, by considering some sample mistakes being detailed in my report.

Report

I found the submitted work enough interesting and appropriate as a contribution to the
SciPost Physics Proceedings. The text is well organized and of general interest, following
some other previous works by the authors.
However, it needs some general text corrections to avoid misprints or language mistakes,
as well as a revision in the presented formalism.

I can make one general remark, which is happening when the authors are citing
references of some of them (not all the authors) by "we...".
See, for example the paragraph starting after Eq.(2).
In such a case, it will be more appropriate to use the passive voice (non-personal) format:
Instead "... we proposed the GPT potential,...", better replace by "...the GPT potential was proposed,... ".

In the following, I am including other specific comments with recommendations, which are indicating that a
throughout revision is necessary, to improve the manuscript presentation.

Abstract

As considering that the abstract is usually being used to advertise a work, even before the
interested reader can access the full paper, I have the following suggestions:

- To define the acronymous GPT in the first time that is appearing in the abstract.
- Revise the abstract in a succinct way, avoiding details on specific reactions (which require
further explanations or definitions that are not easy to be clearly included in an abstract).
- One sentence may be included on the motivation to investigate interactions for
ultra-low energy nuclear synthesis.

Sections 1-4

- Chemical elements should be clearly defined, when appearing for the
first time, to be comprehensive to a general reader. Without clear definitions, one
could be confused with the use of symbols such as D (deuterium atom) and d
(deuteron nucleus) along the text.
- "Pd-cage" should also be explained in a short way.
- Second paragraph after eq.(2), punctuation are required in the starting sentence:
"Let us recall our theory briefly at the Q2T. The Born...".
- Eq.(3) needs to be corrected: $Z$ is defined in the lhs as a function of $q,q^\prime$.
But in the rhs, we have $p$, $p^\prime$ and $q$. The authors should also revise some other
expressions following that; as well as vector notations wherever are necessary.
- The statement, in the last phrase of section 4 (conclusion and discussion) that their
calculation is done at "room temperature nuclear synthesis based upon the Faddeev approach",
needs to be clarified (or removed).
Where "temperature" is introduced (and being effective) in their formulation or calculations?

Formalism

- The authors need to revise the formalism for the notations, considering that all the vector positions (and
corresponding vector momenta) are expressed as being non-vectors. For that, they may follow some
of their previous works.
- Before the Eqs.(16) - (19), they need to define $V_e\equiv V_e(\vec{r}_1,\vec{r}_2,\vec{r}_3)$.
- For clarity on the conditions of (16) and (17), better say (for $\vec{r}_{ki}=\vec{r}_{jk}=0$) in (16);
and [for $\vec{r}_{ki}=0$ ($\vec{r}_{jk}\ne 0$) or $\vec{r}_{jk}= 0$ ($\vec{r}_{ki}\ne 0$)] in (17).
- In Eq.(20), it should be $3\cos^2\theta_k$ instead of $2\cos^2\theta_k$.

References

- Include details of Ref.[6], if available.
- Ref.[10]: replace "Phys. Rev. {\bf 76}" by "Phys. Rev. C{\bf 76}".
- Ref.[14]: "T-matrix", instead of "T-matarix".

Requested changes

The requested changes are described in attached report. Within the recommendations enumerated, particular attention should be given to the formalism.

---

## Round 2 · Author Response

We rewrite our paper by refree comments.

---

## Round 2 · List of Changes

Abstract
As considering that the abstract is usually being used to advertise a work, even before the interested reader can access the full paper, I have the following suggestions:
- To define the acronymous GPT in the first time that is appearing in the abstract.
Ans. Rewritten.

- Revise the abstract in a succinct way, avoiding details on specific reactions (which require further explanations or definitions that are not easy to be clearly included in an abstract).
Ans. Rewritten.

- One sentence may be included on the motivation to investigate interactions for
ultra-low energy nuclear synthesis.
Ans. Rewritten.

Sections 1-4
- Chemical elements should be clearly defined, when appearing for the
first time, to be comprehensive to a general reader. Without clear definitions, one
could be confused with the use of symbols such as D (deuterium atom) and d
(deuteron nucleus) along the text.
Ans. Being explained.

- "Pd-cage" should also be explained in a short way.
Ans. Being explained.

- Second paragraph after eq.(2), punctuation are required in the starting sentence:
"Let us recall our theory briefly at the Q2T. The Born...".
Ans. corrected.

- Eq.(3) needs to be corrected: ZZ is defined in the lhs as a function of q,q′q,q′ .
But in the rhs, we have pp , p′p′ and qq . The authors should also revise some other
expressions following that; as well as vector notations wherever are necessary.
Ans. Vector notation is given.

- The statement, in the last phrase of section 4 (conclusion and discussion) that their
calculation is done at "room temperature nuclear synthesis based upon the Faddeev approach", needs to be clarified (or removed).
Where "temperature" is introduced (and being effective) in their formulation or calculations?
Ans. The usual fusion is performed in the high temperature in the pile, however, in the present theme, the room temperature is commonly used in the technology field. However, in physics, the temperature should be defined by the Boltzmann distribution function. In order to concentrate our discussion, we don’t touch with the detail, we have used a word “the ultralow energy”, but partly we used room temperature without notice. By the referee’s suggestion, the word should be unified in the physics stile. In my Surry talk, we compared the different temperature cases for T=10^9K, T=10^6K, and T=500K. In this paper, the discussions were not mentioned about the temperature but the ultralow energy region. Thanks to the referee!!

Formalism
- The authors need to revise the formalism for the notations, considering that all the vector positions (and corresponding vector momenta) are expressed as being non-vectors. For that, they may follow some of their previous works.
Ans. Corrected.

- Before the Eqs.(16) - (19), they need to define Ve≡Ve(→r1,→r2,→r3)Ve≡Ve(r→1,r→2,r→3) .
- For clarity on the conditions of (16) and (17), better say (for →rki=→rjk=0r→ki=r→jk=0 ) in (16);
and [for →rki=0r→ki=0 (→rjk≠0r→jk≠0 ) or →rjk=0r→jk=0 (→rki≠0r→ki≠0 )] in (17).
Ans. Corrected.

- In Eq.(20), it should be 3cos2θk3cos2⁡θk instead of 2cos2θk2cos2⁡θk .
Ans. Corrected.

References
- Include details of Ref.[6], if available.
- Ref.[10]: replace "Phys. Rev. {\bf 76}" by "Phys. Rev. C{\bf 76}".
- Ref.[14]: "T-matrix", instead of "T-matarix".
Ans. Corrected.

---

## Editorial Decision

published